# Spatiotemporal correlation of spinal network dynamics underlying spasms in chronic spinalized mice

Carmelo Bellardita*, Vittorio Caggiano[†‡], Roberto Leiras[†], Vanessa Caldeira, Andrea Fuchs, Julien Bouvier[§], Peter Löw, Ole Kiehn*

Mammalian locomotor Laboratory, Department of Neuroscience, Karolinska Institutet, Stockholm, Sweden

**Abstract** Spasms after spinal cord injury (SCI) are debilitating involuntary muscle contractions that have been associated with increased motor neuron excitability and decreased inhibition. However, whether spasms involve activation of premotor spinal excitatory neuronal circuits is unknown. Here we use mouse genetics, electrophysiology, imaging and optogenetics to directly target major classes of spinal interneurons as well as motor neurons during spasms in a mouse model of chronic SCI. We find that assemblies of excitatory spinal interneurons are recruited by sensory input into functional circuits to generate persistent neural activity, which interacts with both the graded expression of plateau potentials in motor neurons to generate spasms, and inhibitory interneurons to curtail them. Our study reveals hitherto unrecognized neuronal mechanisms for the generation of persistent neural activity under pathophysiological conditions, opening up new targets for treatment of muscle spasms after SCI.

**\*For correspondence:** carmelo.bellardita@ki.se (CB); Ole.Kiehn@ki.se (OK)

[†]These authors contributed equally to this work

**Present address:** [‡]Computational Biology Center, IBM TJ Watson Research Center, New York, United States; [§]Paris Saclay Institute of Neuroscience, UMR9197 – CNRS and Universite Paris-Sud, Gif-sur-Yvette, France

## Introduction

Spinal cord injury (SCI) leads to disturbed sensory, autonomic, and motor functions. In the immediate phase following SCI the excitability of the spinal networks caudal to the injury is completely depressed (*Klussmann and Martin-Villalba, 2005*; *Rossignol et al., 2007*; *Schwab et al., 2006*). This initial state of motor depression is often followed by a maladaptive state of hyper-excitability of the spinal circuitries referred to as spasticity (*Biering-Sørensen et al., 2006*; *Dietz, 2010*; *Frigon and Rossignol, 2006*; *Hultborn, 2003*; *Little et al., 1999*; *Nielsen et al., 1995*, *2007*) and characterized by an increase in spinal reflexes and by the appearance of involuntary sustained muscle contractions or spasms (*Nielsen et al., 2007*) often triggered by innocuous sensory stimuli or muscle stretch.

At the premotor neuron level appearance of spasms has been hypothesized to be related to an increased gain in sensory signaling to motor neurons caused by a reduced pre-synaptic inhibition of sensory afferents (*Faist et al., 1994*; *Xia and Rymer, 2005*), an alteration in the efficacy of spinal inhibitory circuits (*Crone et al., 1994*, *2004*; *Pierrot-Deseilligny et al., 1979*; *Shefner et al., 1992*), or a shift from a hyperpolarizing to a depolarizing action of the inhibitory synaptic inputs onto motor neurons (*Boulenguez et al., 2010*). At the motor neuron level the development of spasms has been linked to the increased expression of plateau potentials – carried by voltage-gated persistent $Ca^{2+}$ and $Na^+$ currents – allowing motor neurons to sustain tonic firing (*Bennett et al., 1999*; *Eken et al., 1989*; *Kiehn and Eken, 1998*; *Li and Bennett, 2003*; *Li et al., 2004a*; *Murray et al., 2010*). However, all behaviorally relevant motor activities require excitatory drive to orchestrate the generation, maintenance and termination of motor responses (*Hägglund et al., 2010*; *Kiehn, 2016*), leading to the possibility that muscle spasms may be the result of aberrant spinal excitatory drive to motor

neurons. Indeed long-lasting excitatory postsynaptic potentials may be detected in motor neurons during sustained motor responses in chronic spinalized animals (*Akay et al., 2014*; *Brumovsky, 2013*). Nevertheless, whether activity in interneurons is involved in initiating, maintaining, and terminating spasms is unknown.

To address these issues we have developed a chronic spinal cord mouse model that mimics aspects of SCI and offers the possibility to combine detailed electrophysiological and imaging studies of both motor neurons and neurotransmitter-defined populations of interneurons to directly monitor and perturb their activity during muscle spasms. The approach builds on a chronic spinal cord model developed in the rat (*Bennett et al., 1999*) that targets a transection to the sacral segment of the spinal cord thereby producing spasms in the tail muscles without affecting limb muscles or bowel and bladder functions. We find an increased but graded expression of plateau potentials in motor neurons, which alone does not account for spontaneous and sensory-evoked spasms. However, optical imaging in chronically spinal cord transected mice unmasked persistent neural activity in spinal interneurons with distinct spatiotemporal relationship during muscle spasms. Optogenetic activation and inactivation of spinal excitatory interneurons causally linked their activity to the triggering and maintenance of the spasms. In contrast, optogenetic stimulation of spinal inhibitory interneurons revealed a functional inhibition that played a role in sculpturing and curtailing muscle spasms.

Our study reveals a hitherto unrecognized mechanism for muscle spasms after chronic spinal cord transection whereby assemblies of excitatory interneurons are recruited into functional circuits by weak sensory inputs, generating persistent excitatory activity that drives motor neurons even if inhibition is still functionally active. Accordingly, our study shifts the widely held view that muscle spasms after SCI are the result of increased motor neuron excitability and reduced spinal inhibition acting as a source of excitation. It therefore provides new directions for treatment of spasms after SCI.

## Results

### Appearance of spasms in tail muscles after chronic spinal cord transection in mice

We generated a mouse model of SCI with a transection at the second segment of the sacral spinal cord, affecting just tail muscles. Spasticity developed gradually in the tail, plateauing eight weeks after transection and a chronic change with a typical phenotypical hook-shape appeared in tail posture (bottom panel in *Figure 1A* and *Video 1*). Spontaneous muscle spasms slowly emerged, leading to an increased and prolonged bending of the tail (*Figure 1B* and *Video 2*).

To evaluate the muscle activity underlying spasms, we recorded activity of multiple motor units simultaneously in different segments along the tail during spontaneous spasms (*Figure 1C* and *Video 2*). Usually a spasm was preceded by low or no motor unit activity (grey box, *Figure 1D*) and initiated by an abrupt activation of several motor units in different segments of the tail (green box, *Figure 1D*). The massive muscle contraction could last up to several minutes before it was slowly released (violet box *Figure 1D*). Spontaneous spasms occurred on a regular base in the chronic phase after spinal cord transection (*Figure 1E*) with a main rostro-caudal recruitment of the motor units (*Figure 1F*), even though caudo-rostral recruitment of the motor units may also appear.

Gentle stroking of the tail skin (*Video 2*), presumably activating low threshold mechanoceptors or stretch-activated muscle spindles, could initiate prolonged spasms similar to those spontaneously evoked, demonstrating that pronounced triggers for spasms are sensory stimuli. Similarly, low intensity electrical stimulation of the tail nerves could evoke prolonged spasms (data not shown).

Together, these data demonstrate that the chronic sacral transection in adult mouse provides a valid model for studies of spasm development after SCI.

### Motor neurons and interneuron circuits develop hyper-excitability after chronic spinal cord transection

To systematically evaluate the motor neuron and network excitability after chronic spinal transection we used electrical stimulation to evoke spasms. We found a decreased threshold for the

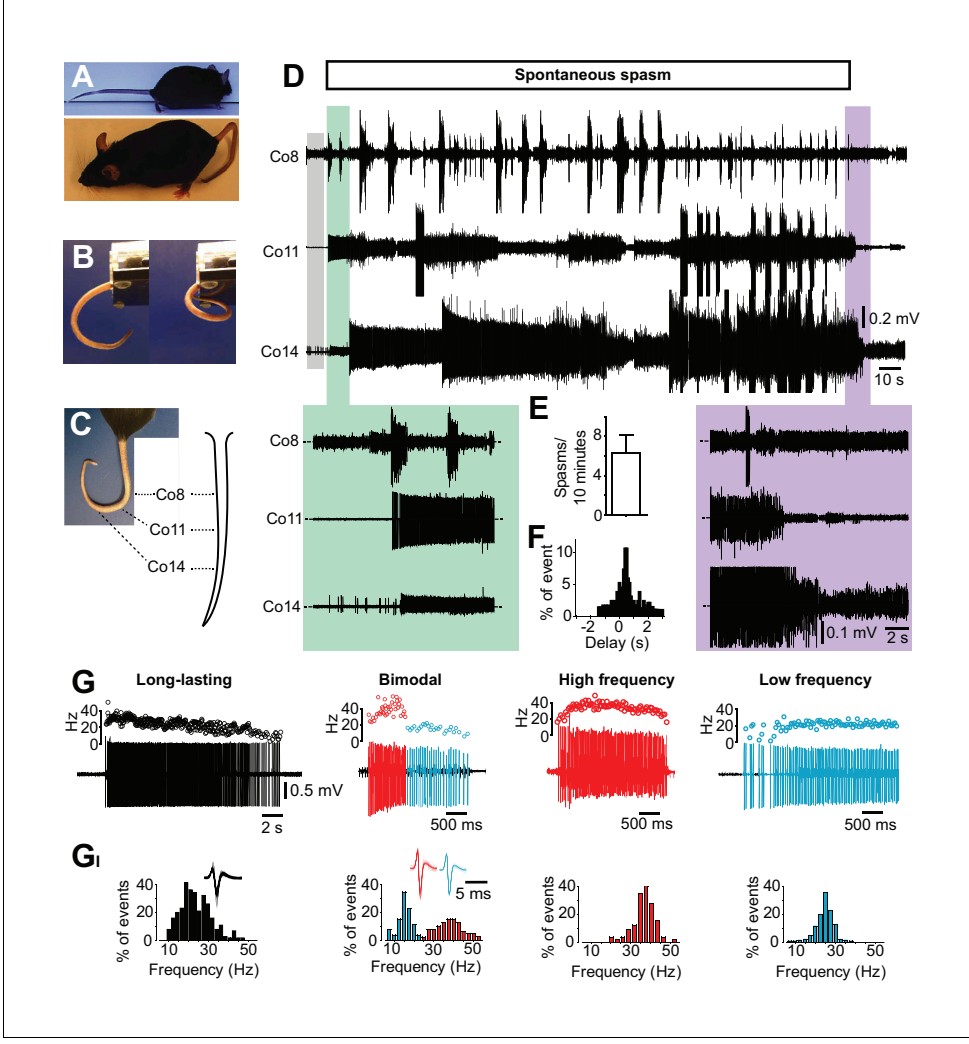

**Figure 1.** Occurrence of spasms in a mouse model of spinal cord injury affecting just tail muscles. (**A**) Resting position in acutely (upper picture) or chronically S2-transected (lower picture) mice. (**B**) Tail position in S2-transected mouse before (left) and during (right) the occurrence of a spontaneous spasm (see also *Video 2*). (**C**) Positions of the recording electrodes for EMGs along the tail. (**D**) Gross EMG recordings in three different segments of the tail (Co8, Co11 and Co14) in a chronically S2-transected mouse during a spasm. A period of no or little EMG activity (grey box) preceded the prolonged activation of motor units observed in different segments of the tail ('on episode', expanded in green box). The activity propagated in a proximal distal direction before it spontaneously terminates ('off episode' expanded in violet box). (**E**) Number of muscle spasms in S2-transected animals two months after transection (N = 5). (**F**) Frequency distribution histograms of time delay (s) between activation of rostral and caudal motor units (n = 46). (**G**) Four phenotypic firing profiles of motor units – long lasting, bimodal, high frequency, and low frequency – triggered during spontaneous spasms. Raw recordings and instantaneous firing frequencies. (**G_I**) Respective frequency distribution histograms of the firing frequencies for motor units in **G**.

The following source data and figure supplements are available for figure 1:

**Source data 1.** Related to *Figure 1*.
**Source data 2.** Related to *Figure 1*.
**Figure supplement 1.** Motor neuron and interneuron hyper-excitability after chronic spinal cord transection assessed by reflex responses and prolonged firing.
**Figure supplement 1—source data 1.** Related to *Figure 1—figure supplement 1*.

*Figure 1 continued on next page*

*Figure 1 continued*

**Figure supplement 1—source data 2.** Related to *Figure 1—figure supplement 1*.

**Figure supplement 1—source data 3.** Related to *Figure 1—figure supplement 1*.

monosynaptic activation of motor neurons by low threshold afferents after the chronic transection as compared to acute transection (defined as control, *Figure 1—figure supplement 1A–B*), suggesting increased motor neuron excitability. Moreover, prolonged muscle activity could be induced by electrical stimulation subthreshold for the monosynaptic activation of motor neurons (the H-reflex) with a different time course from high-threshold stimulation (*Figure 1—figure supplement 1A–B*). Typically, with low-threshold stimulation there was a delay in spasm's development with a silent period between the time of stimulation and the prolonged motor response (*Figure 1—figure supplement 1C–D*).

The delayed development of the spasm with low-threshold stimulation shows that spasms can be triggered without a direct monosynaptic input onto motor neurons from sensory afferents, pointing to involvement of spinal interneurons in spasm generation. We therefore evaluated the motor neuron and interneuron hyper-excitability separately.

## Motor units display four distinct firing patterns after chronic spinal cord transection that are uniquely related to variable expression of plateau properties

To address the role of motor neurons in spasm generation, we first recorded single motor unit activity during spontaneous spasms. We identified four main phenotypic firing patterns (*Figure 1G–G₁*): long-lasting firing with a gradual decrease in instantaneous firing frequency (27 ± 6.3, n = 25, N = 5); a bimodal firing, characterized by an initial phase with a high and unstable instantaneous firing frequency (41.48 ± 13.15 Hz, n = 28, N = 5) followed by a lower and more stable firing frequency (23.85 ± 8.5 Hz); and firing where only high (45.55 ± 13.64, n = 9, N = 5) or low firing frequencies (17.87 ± 2.5 Hz, n = 11, N = 5) were expressed. The phenotypic firing patterns of the motor units could also be induced by gentle stroking of the tail (*Figure 1—figure supplement 1E*) or low-intensity electrical stimulation of the tail nerves (*Figure 1—figure supplement 1F*). There was no correlation between the different motor unit types and their specific distribution along the rostro-caudal axes of the tail.

These data show that the individual motor unit contributes with variable firing patterns to prolonged spasms.

To test if firing behaviors of the motor units correlate with the expression of motor neuron properties - in particular plateau properties - we used the isolated sacral spinal cord preparation from acutely and chronically spinal cord transected animals. The isolated spinal cord of chronically transected mice showed all the characteristics of the in situ conditions, including spontaneous (*Figure 2—figure supplement 1A–B*) or dorsal root stimulus-induced prolonged motor activity (spasms) mimicking sensory activation in vivo (*Figure 2—figure supplement 1C-D*), increased motor neuron and interneuron excitability *(Figure 2—figure supplement 1E-F*).

To test whether the four different firing patterns result from the expression of motor neuron plateau potential properties (*Conway et al., 1988*; *Heckmann et al., 2005*;

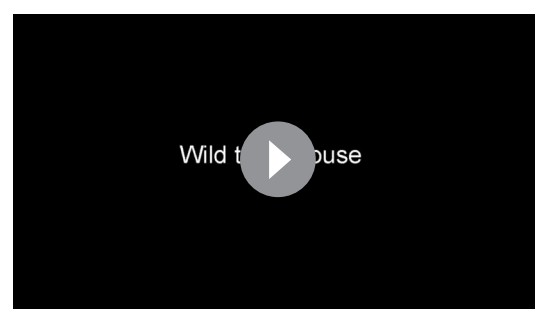

**Video 1.** Related to *Figure 1*. Tail posture in acutely and chronically spinal cord transected mice. The Movie shows the natural posture of the tail during standing and over-ground locomotion in a acutely transected mouse and in a mouse two months after transection of the second segment of the sacral spinal cord.

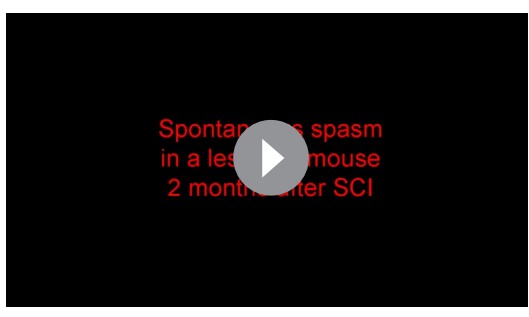

**Video 2.** Related to *Figure 1*. Spontaneous and light sensory-evoked spasms in chronically spinal cord transected mouse. The movie shows chronically S2-transected mouse in a restrainer with the tail free to move. A spontaneous spasm that curved the entire tail was followed by a spasm evoked by gentle stroking of the tail skin.

*Hounsgaard et al., 1988*; *Kiehn and Eken, 1998*), shown to contribute to prolonged firing in chronic spinalized rats (*Bennett et al., 1999*) and cats (*Eken et al., 1989*), we performed sharp electrode recordings from motor neurons in the isolated mouse spinal cord preparation (*Figure 2—figure supplement 2*). First, we applied low-threshold electrical stimulation to the dorsal root (*Figure 2A–B*). The four phenotypic firing patterns, observed in vivo, were also found in motor neurons from chronically spinal cord transected mice in vitro (*Figure 2C–F*) while the motor neurons of spinal cords from acutely transected mice displayed a stereotypic response with no activity after termination of the stimulation (*Figure 2B*). Then we injected slow triangular current ramps to estimate the presence of plateau properties (*Hounsgaard et al., 1988*). Typically, motor neurons with strong plateau properties show prolonged firing on the descending half of the ramp leading to a counter-clockwise hysteresis in the frequency-current relationship (f-I) (*Hounsgaard et al., 1988*), and recruitment threshold that is more positive than the de-recruitment threshold leading to a negative $\Delta I$ (recruitment-de-recruitment current) (*Bennett et al., 2001b*). Motor neurons with long-lasting discharge induced by dorsal root stimulation showed strong plateau properties with counter-clockwise hysteresis and negative $\Delta I$ ($-1.5 \pm 0.6$ nA, mean $\pm$ SD, n = 7, N = 8, *Figure 2C*). We refer to these cells as 'full plateau motor neurons'. Motor neurons with a bimodal firing pattern showed linear f-I plots but asymmetric firing with higher recruitment than de-recruitment thresholds ($\Delta I = -0.6 \pm 0.4$ nA n = 32, N = 8, *Figure 2D*). We refer to these neurons as 'Partial plateau motor neurons'. Motor neurons that exhibited short-lasting but high frequency responses displayed a linear or a clockwise hysteresis of the f-I plots with lower recruitment than de-recruitment thresholds ($\Delta I = 168 \pm 60$ pA, n = 6, N = 8, *Figure 2E*). We refer to these motor neurons as 'No plateau motor neurons'. Motor neurons with sustained low frequency response after dorsal root stimulation exhibited intermediate plateau properties ($\Delta I = -1.1 \pm 0.9$ nA, n = 6, N = 8, *Figure 2F*). Most motor neurons recorded in the isolated spinal cord from time-matched sham operated and acutely spinalized animals displayed no sign of plateau potentials (21 out of 25 motor neurons, N = 5).

The chronic spinal cord transection, therefore, induces changes in motor neuron properties with a bias towards the expression of plateau potentials. However, the number of motor neuron expressing full plateau properties that may support sustained firing was only about 10% (*Figure 2G*). Most motor neurons expressed partial plateau properties, which may promote but not sustain prolonged firing. Therefore, while motor neuron plateau potentials may contribute they alone seemed unlikely to account for the orchestrated and sustained motor activity underlying muscle spasms after chronic spinal cord transection. Accordingly, we focused our attention on premotor circuits.

## Activity in excitatory and inhibitory circuits maintains and shapes motor neuron firing during spasms

We first assessed the synaptic inputs to motor neurons during spasms. A sustained barrage of synaptic inputs impinged on motor neurons during muscle spasms when motor neurons were hyperpolarized below their firing threshold (to $-75$ mV, *Figure 3A–B*). The silent period immediately following the termination of the sensory stimulation was dominated by inhibitory post-synaptic potentials that were outnumbered by massive long-lasting excitatory inputs initiating and sustaining the strong depolarization underlying the spasm (*Figure 3C–F*). Both the duration of the silent period and the duration of depolarization underlying spasms were modulated by the stimulation intensity (*Figure 3—figure supplement 1*). Stronger stimulation strength phase advanced the silent period and enhanced the duration and the amplitude of the spasm.

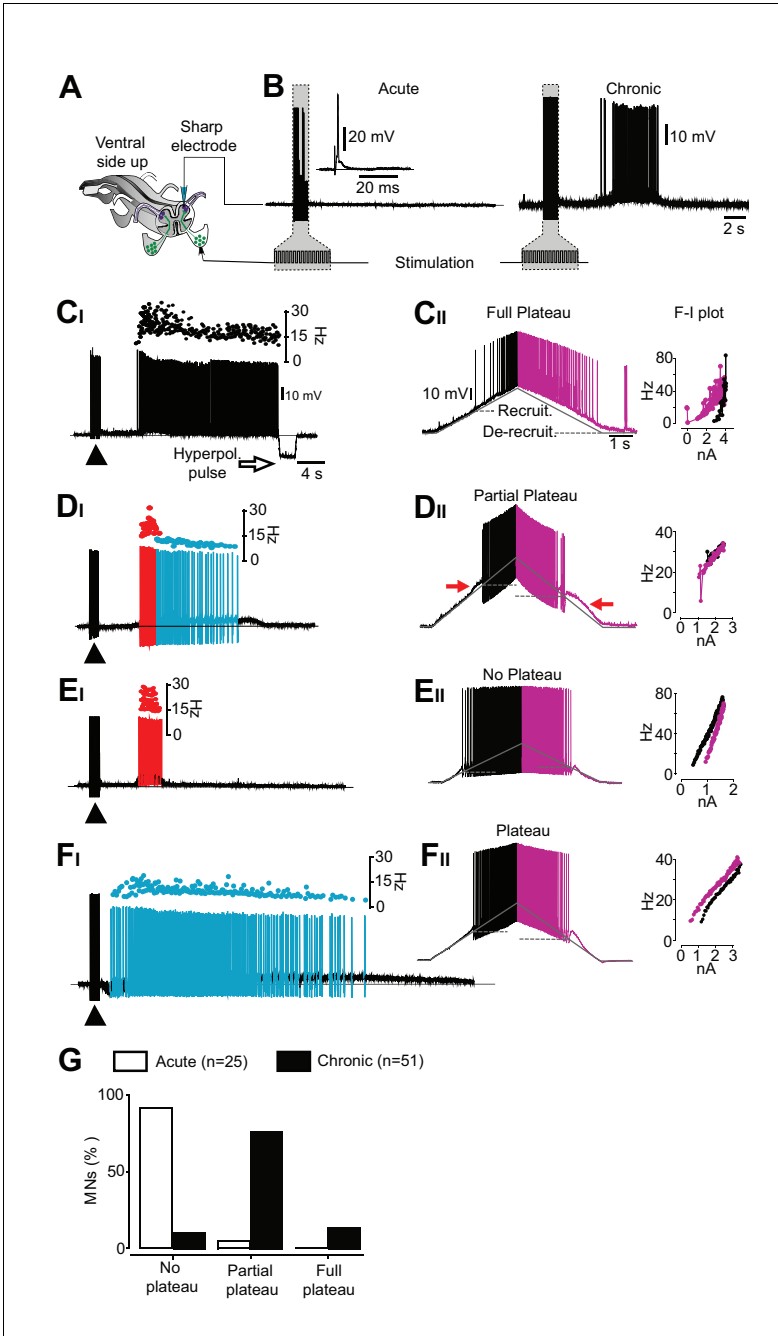

**Figure 2.** An exclusive relationship between firing pattern and expression of plateau properties in motor neurons after chronic spinal cord transection. (A) Schematic of the isolated spinal cord with configuration of electrodes placed on dorsal and ventral roots and intracellular recording of motor neuron. (B) Examples of motor neuron activity in response to dorsal root stimulation in acutely (left trace) and chronically spinal cord transected mice (right trace). The grey box indicates the stimulation. (C–F) Expression of plateau properties as assessed by current ramps. Motor neurons with long-lasting firing after dorsal root stimulation (C$_I$) exhibited strong plateau properties with typical counter-clockwise hysteresis in the f-I plot (C$_{II}$). Motor neurons with bimodal firing patterns (Hartigan dip. test <0.05, D$_I$) expressed partial plateau properties with linear f-I plot and activation and de-activation of the short-lasting plateau below the threshold for the action potentials (D$_{II}$, red arrows). Motor neurons with high frequency firing (E$_I$) exhibited linear f-I plot or clockwise hysteresis in the f-I plot indicative of no plateau potentials (E$_{II}$). Motor neurons with slow firing (F$_I$) may exhibit strong plateau properties and counter-clockwise hysteresis in the F-I plot (F$_{II}$). (G) Summary of the expression of plateau properties in motor neurons in spinal cords from acutely and chronically spinal cord transected mice.

*Figure 2 continued on next page*

*Figure 2 continued*

The following source data and figure supplements are available for figure 2:

**Source data 1.** Related to *Figure 2*.

**Figure supplement 1.** Motor neuron and interneuron hyper-excitability after chronic spinal cord transection assessed in vitro.

**Figure supplement 1—source data 1.** Related to *Figure 2—figure supplement 1*.

**Figure supplement 2.** Motor neuron recordings in sacral spinal cord and their electrophysiological identification.

Further confirmations of premotor circuit involvement in spasm generation emerged from pharmacological dissection of the motor neuron inputs. When inhibitory glycinergic and Gabaergic transmission was blocked, the silent period preceding the spasm in pre-drug conditions disappeared (*Figure 3G*), showing that the silent period in motor neurons was the result of activity in premotor inhibitory interneurons. In contrast the spasm was completely abolished when mephenesin, a blocker of polysynaptic mostly glutamatergic transmission (*Cazalets et al., 1995*; *Kiehn and Butt, 2003*; *Lev-Tov and Pinco, 1992*) was applied (*Figure 3H*). Altogether, these experiments show that the spasms were triggered by polysynaptic glutamatergic pathways, and not from direct afferent inputs to motor neurons.

## Excitatory and inhibitory interneuronal activity increases during spasms after chronic spinal cord transection

To specifically evaluate the contribution of spinal interneurons to muscle spasms we next used genetically driven calcium imaging to separately visualize the activity of excitatory (eINs) and inhibitory (iINs) interneurons during spasms. We crossed *Rosa26-LSL-GCaMP3 (Ai38)* mice with mice expressing Cre under control of the gene *Slc17a6,* that codes *for* the vesicular glutamate transporter 2 (Vglut2) (*Borgius et al., 2010*) or the gene Slc32a1, coding the vesicular inhibitory amino acid transporter (VIAAT) (*Hägglund et al., 2013*) to drive the expression of the calcium-sensitive protein GCaMP3 in excitatory (Vglut2) or inhibitory (VIAAT) spinal neurons respectively (*Figure 4—figure supplement 1* and see *Hägglund et al., 2013*).

By having the transverse section of the cord facing the microscope objective (*Figure 4A*), we first directly visualize the interneuron population responses in the dorsal and ventral horns (*Figure 4B*). In both acutely and chronically transected spinal cords low-threshold stimulation of dorsal roots elicited activity in eINs in the superficial dorsal horn with a peak of activity during the stimulation (*Figure 4C–D*, *Video 3*). However in chronically spinal cord transected mice the initial activity was followed by a second wave of excitatory activity, greater in intensity and longer in duration. It appeared in the ventro-medial horn of the spinal cord and spread to the dorsal horn, finally peaking 3–4 s after the end of the stimulation (red trace in the lower panel of *Figure 4D*). Lastly a third wave of excitatory activity appeared again in the deep ventral horn, corresponding to the activity observed in the ventral root recordings (blue trace in the lower panel of *Figure 4D*). This sequence of events was observed in all chronically spinal cord transected animals (*Figure 4E*, N = 5) and it directly revealed the spatial location and the temporal dynamics of the excitatory circuit activity in response to sensory stimulation in chronically transected mice before, during and after spasms.

A change in activity pattern was also seen in iINs when acutely and chronically spinal cord transected mice were compared (*Video 4*). In acutely spinal cord transected mice dorsal root stimulation evoked a short-lasting response of iINs in the dorsal horn (*Figure 4F*, red trace lower panel). Surprisingly, in chronically spinal cord transected mice the same stimulation resulted in a greater and sustained activation of iINs whose activity spreads from the dorsal horn (red trace in *Figure 4G*) to the ventral horn, lasting several seconds (blue trace in *Figure 4G*). Thus contrary to expected, inhibitory circuits exhibited a greater and sustained activation after chronic spinal cord transection, suggesting their involvement in muscle spasm sculpturing and termination.

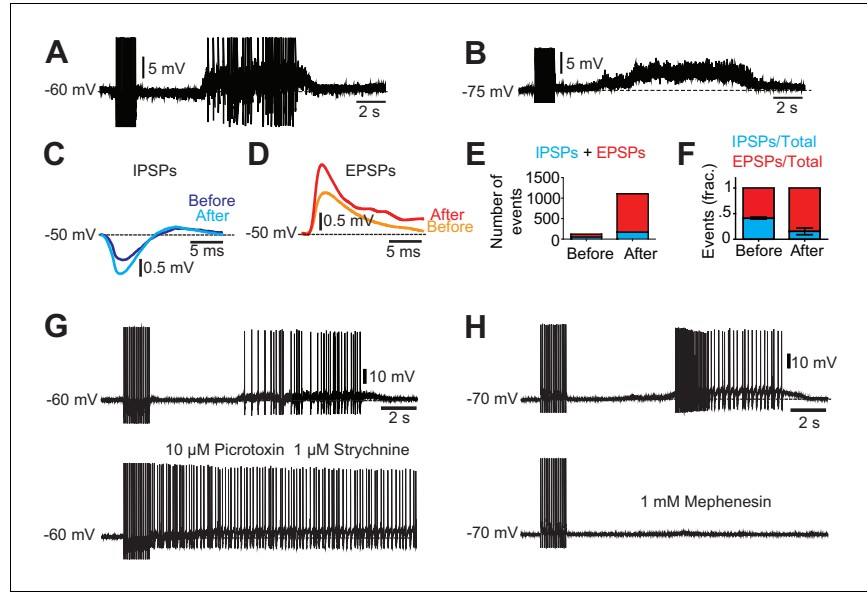

**Figure 3.** Excitatory and inhibitory inputs onto motor neurons after chronic spinal cord transection. (**A**– **B**) Motor neuron response to low-threshold dorsal root stimulation (DRS) at resting membrane potential (**A**) and when hyperpolarized (**B**). Note the prolonged barrage of synaptic potentials. (**C–D**) Synaptic events divided in inhibitory post-synaptic potentials (IPSPs) and excitatory post-synaptic potentials (EPSPs) when the motor neuron was depolarized to −50 mV. Composite IPSPs and EPSPs before DRS (blue and orange respectively) were smaller than after DRS (cyan and red). Mean traces from all events before and after stimulation. (**E**) Number of events (IPSPs in blue and EPSPs in red) before and after DRS in a selected motor neuron. (**F**) Relative fraction of IPSPs and EPSPs before and after DRS (n = 10, N = 3). The relation between excitatory and inhibitory inputs shifted from being equally balanced before DRS to being dominated by excitation after DRS. Note that after DRS the synaptic events increases in number and in synchrony, allowing for summation. (**G**) The silent period following stimulation and preceding the spasm disappeared when GABAergic and glycinergic inhibitory circuits were blocked with picrotoxin (10 µM) and strychnine (1 µM, six out of six cells) (upper trace pre-drug). (**H**) Firing of a motor neuron in a chronically transected spinal cord before (upper trace) and after 1 mM Mephenesin (middle trace) which blocks polysynaptic transmission After Mephenesin, the spasm was severely reduced and was only revealed as a curtailed version when the cell was depolarized (lower trace). The direct activation of motor neurons from dorsal root stimulation was preserved. These effects were seen in all motor neurons tested (five out of five cells).

The following source data and figure supplements are available for figure 3:

**Source data 1.** Related to *Figure 3*.

**Source data 2.** Related to *Figure 3*.

**Figure supplement 1.** The duration of the silence period and the spasms are modulated by the stimulation intensity.

**Figure supplement 1—source data 1.** Related to *Figure 3—figure supplement 1*.

**Figure supplement 1—source data 2.** Related to *Figure 3—figure supplement 1*.

These experiments show that excitatory and inhibitory circuits located both in the dorsal and ventral horns show prolonged activity after a chronic spinal cord transection and their activity is temporally related to the spasms.

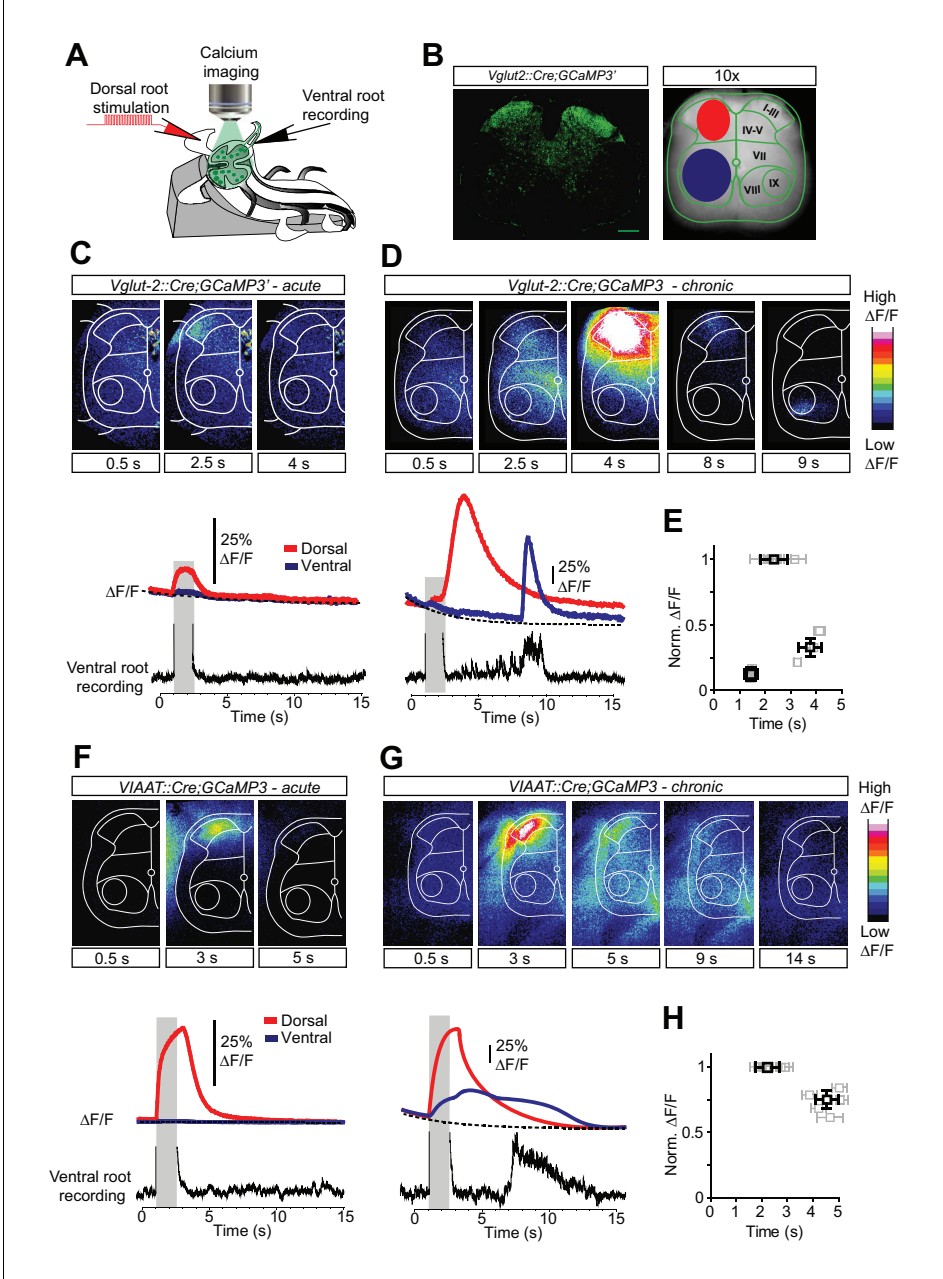

**Figure 4.** Increased activity in spinal interneurons after chronic spinal cord transection. (A) Schematic of the set-up for calcium imaging. (B) Confocal image of a transverse section of a sacral spinal cord from $Vglut2^{Cre}$;R26-GCaMP3 (left panel) and a snapshot of the spinal cord as seen with a 10x objective with the main areas of interest for detecting change in fluorescence indicated(dorsal horn in red and ventral horn in blue, right panel). (C–H) Spatial and temporal development of changes in fluorescence intensity in excitatory (C–E; $Vglut2^{Cre}$;R26-GCaMP3) and inhibitory (F–H; $VIAAT^{Cre}$;R26-GCaMP3) spinal neurons in acutely (C, F) and chronically spinal cord transected (D, G) mice. Upper panels show frames of fluorescence transients ($\Delta$F/F) at specific time points after low-threshold dorsal root stimulation. Lower panels show change in fluorescence ($\Delta$F/F) as a function of time in the dorsal (red) and ventral (blue) horns with the corresponding rectified ventral root recording. Grey bars represent stimulation. The plots summarize the time and the normalized intensity of the peaks for the eINs (E) and iINs (H) after acute and chronic transections (mean ± SD in black, N = 5 for each group).

The following source data and figure supplements are available for figure 4:

**Source data 1.** Related to *Figure 4*.

*Figure 4 continued on next page*

*Figure 4 continued*

**Source data 2.** Related to *Figure 4*.
**Figure supplement 1.** Specific expression of GCaMP3 in *Vglut2^Cre* and *VIAAT^Cre* mice.
**Figure supplement 1—source data 1.** Related to *Figure 4—figure supplement 1*.

## Sequential activation of eINs, iINs and MNs during sustained motor activity after chronic spinal cord transection

To reveal the spatio-temporal organization of spinal INs activity in relation to the motor response, we subsequently imaged single cells in dorsal, intermediate, and ventral areas of the spinal cord (*Figure 5A*). EINs and iINs in chronically spinal cord transected mice exhibited spatially and temporally segregated activity (*Figure 5B*). In order to reveal the temporal relationship between the activity of eINs, iINs and motor neurons in the same segment, the response of the INs was normalized to the ventral root response which was divided in four periods: stimulation period (Stim; from −1 to 0), pre-spasm silence (Sil; from 0 to 1), spasm (spasm; from 1 to 2), and post-spasm silence (post-sp; from 2 to 3). In spinal cords of acutely spinal cord transected mice, low-threshold stimulation of the dorsal root generated a sequential activation of eINs with a mean peak of activity during the stimulation period (−0.22 ± 0.24, mean ± SD) followed by iINs immediately after the end of the stimulation (0.06 ± 0.13). A similar sequential activation with a first peak of excitatory activity during the stimulation (−0.33 ± 0.07) followed by inhibitory activity occurring in the silent pre-spasm period (0.17 ± 0.19) was present in spinal cords from chronically spinal cord transected mice. However, in chronically spinal cord transected mice, a second delayed peak of excitatory activity appeared at the beginning of the spasm (1.17 ± 0.2) and a second peak of inhibitory activity appeared just before the spasm ended (1.855 ± 0.25, *Figure 5C*).

These experiments show that the spinal network displays persistent neural activity. When this activity is deconstructed into that of individual excitatory and inhibitory neurons and resampled as a population response, it reveals a complex pattern of activity that exhibits a specific relationship between the activation of the eINs and the activity of both the motor pools as well as the iINs whose activity, in contrast, is related to the appearance of the silent period.

## Spinal eINs generate persistent neural activity to drive muscle spasms after chronic spinal cord transection

However, the correlated IN activity does not allow us to determine whether an interneuron population activity is causally related to spasm generation, maintenance, and termination. To provide such insight we expressed the light-activated Channelrhodopsin (ChR2) or Halorhodopsin (*eNpHR3*) in eINs and iINs to allow for their activation or inactivation (*Hägglund et al., 2010, 2013*). We first activated Vglut2^+ cells in vivo with an optical fiber implanted at the level of the first coccygeal segment of the spinal cord (*Caggiano et al., 2014*) (*Figure 6A*). In acutely transected mice (0–7 days) light stimulation generated movements of the tail for the duration of stimulation only (black trace in *Figure 6B*, *Video 5*; N = 5). In contrast, the same stimulation applied to chronically transected mice resulted in a long lasting bending of the tail that

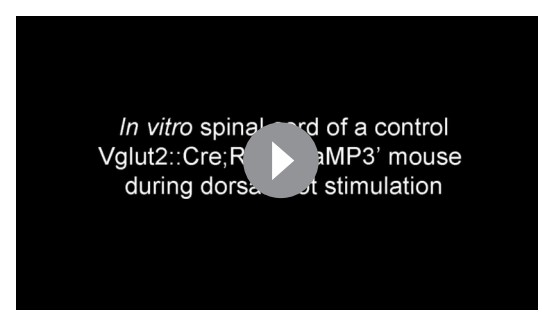

*In vitro* spinal cord of a control Vglut2::Cre;R26-GCaMP3' mouse during dorsal root stimulation

**Video 3.** Related to *Figure 4*. Delayed activation of excitatory spinal interneurons during sensory-evoked spasm after chronic spinal cord transection. The movie shows the calcium response (ΔF/F) in excitatory spinal interneurons (marked in *Vglut2^Cre;R26-GCaMP3* animal) evoked by low threshold dorsal root stimulation in cords from acutely and chronically S2-transected mice. The transverse plane of a hemisection of the isolated sacral spinal cord was acquired with a 10x objective (see *Figure 3a* for a schematic of the experiment). Frequency of acquisition: 20 frame/s.

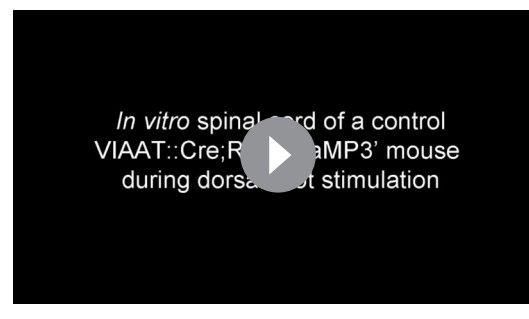

**Video 4.** Related to *Figure 4*. Inhibitory spinal interneurons are strongly activated during spasms after chronic spinal cord transection. The movie shows the calcium response (ΔF/F) in inhibitory spinal interneurons (*VIAAT^Cre^;R26-GCaMP3*) evoked by low threshold dorsal root stimulation in cords from acutely and chronically S2-transected mice. The transverse plane of a hemisection of the isolated sacral spinal cord was acquired with a 10x objective. Frequency of acquisition: 20 frames/s.

lasted several minutes after the end of the stimulation (red trace in *Figure 6B* and *Video 5*; N = 7). Similar prolonged firing was evoked in the isolated sacral cord from chronically spinal cord transected mice when any one of the spinal segments was optogenetically stimulated (*Figure 6C*). The sustained activity lasted even longer when inhibitory transmission was blocked. In contrast it was almost abolished when glutamatergic neuro-transmission was silenced (*Figure 6c*), leaving only a small direct activation of MNs, likely due to early expression of Vglut2 in some MNs (*Nishimaru et al., 2005*).

These experiments suggest that activation of spinal excitatory neurons may drive spasms. However, since the trans-genetic approach is likely to labeled Vglut2 expressing afferents (*Brumovsky, 2013*), there is a possibility that such terminals will also be stimulated by the light, possibly contributing to the activation of intraspinal circuits. To evaluate this possibility we recorded the dorsal root potentials both in acutely and chronically spinalized mice. These recordings showed that light stimulation indeed led to antidromic activation of afferents, but that this activation was short-lasting (20–50 ms) and only present in the beginning of the stimulation

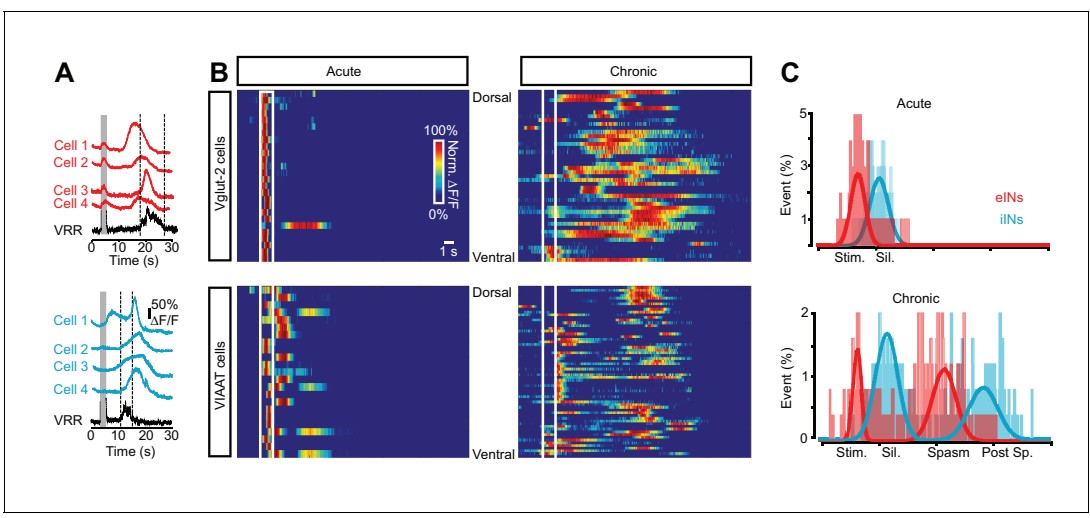

**Figure 5.** Sequential activation of spinal neurons generates persistent neural activity during spasms after chronic spinal cord transection. **(A)** Example of Ca$^{2+}$ signals (ΔF/F) of different cells and the ventral root responses following dorsal root stimulation in Vglut2$^+$ (red traces, upper panel) and VIAAT$^+$ (blue traces, lower panel) cords of chronically spinal cord transected mice. **(B)** Color raster plot of normalized Ca$^{2+}$ response following dorsal root stimulation of excitatory (eINs - upper panels) and inhibitory interneurons (iINs - lower panels) from acutely (left panels) and chronically spinal cord transected (right panels) mice, organized with respect to their dorso-ventral position. **(C)** Compiled frequency distribution of the timing of the peak of calcium activity for eINs (red) and iINs (blue) normalized to the ventral root recordings for acutely (eINs = 98, N = 5; iINs = 98, N = 5) and chronically spinal cord transected mice (eINs = 198, N = 5; iINs = 184, N = 5). Data were fitted with Gaussian distributions and compared using the Student's t test. The x-axes indicate the normalized time: period of stimulation (Stim.), silent period following stimulation (Sil.), period of prolonged motor response (spasm), and period following the spasm (post-sp.). See Materials and methods for information about the time normalization.
The following source data is available for figure 5:

**Source data 1.** Related to *Figure 5*.

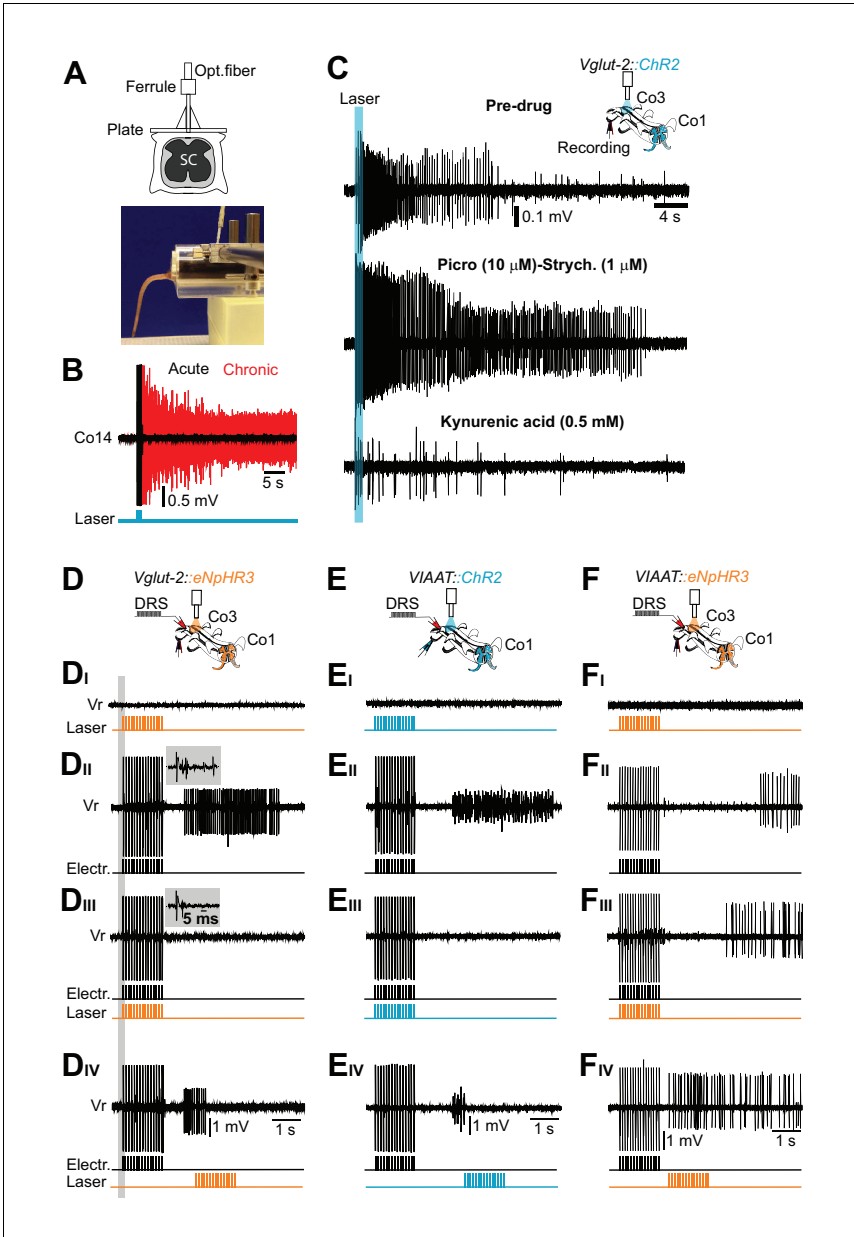

**Figure 6.** Excitatory, but not inhibitory, interneurons trigger the persistent neural activity underlying spasms after chronic spinal cord transection. (**A**) Drawing of the optical fiber implant above the sacral spinal cord (upper panel). (**B**) Gross EMG recordings from tail muscles in acutely (black trace) and chronically spinalized (red trace) *Vglut2^{Cre}; R26ChR2* mice during light stimulation of the first coccygeal segment of the spinal cord (train of 15 pulses, 20 ms, 10 Hz). Stimulation of excitatory interneurons (eINs) in chronically spinalized mice always generated spasms (4 out of 4). See also *Video 4*. (**C**) Co3 ventral root recordings in isolated spinal cord from a chronically spinal cord transected *Vglut2^{Cre}; R26ChR2* mouse during optogenetic stimulation at the third coccygeal segment of the spinal cord (train of 15 pulses, 20 ms, 10 Hz, upper trace), after application of 10 μM of the GABA_A antagonist picrotoxin in combination with 1 μM of the glycinergic antagonist strychnine (middle trace), and after application of a broad range glutamatergic antagonist, Kynurenic acid (0.5 mM, lower trace). (**D–F**) Schematics of the in vitro experiments with indication of dorsal root stimulation (DRS), ventral rootlets recordings and optical stimulation in chronically spinal cord transected *Vglut2^{Cre}; eNpHR3* (D, N = 5), *VIAAT^{Cre}; R26ChR2* (E, N = 10), *and VIAAT^{Cre}; eNpHR3* (F, N = 10) mice. Raw ventral root recordings with only optogenetic stimulation (DI–FI), only electric stimulation (DII–DII), simultaneous optogenetic and electric stimulation (DIII–FIII) and optogenetic stimulation delayed in time (D_{IV}-F_{IV}). Note that optogenetic activation of iINs (E_I) does not evoke spasms while their activation during a spasm immediately terminates it (E_{III}). Insets in D_{II–III} show that the monosynaptic and

*Figure 6 continued on next page*

*Figure 6 continued*

polysynaptic inputs to motor neurons are similar with and without optogenetic stimulation during the electrical stimulation.

The following source data and figure supplements are available for figure 6:
**Figure supplement 1.** Optogenetic stimulation of *Vglut2$^{Cre}$; R26ChR2* mice generates short-lasting but not persistent activity in primary afferents.
**Figure supplement 1—source data 1.** Related to *Figure 6—figure supplement 1*.
**Figure supplement 2.** In vitro optogenetic inhibition of *Vglu2$^{Cre}$; R26eNpHR3* reveals intersegmental excitation supporting persistent neural activity underlying spasms.
**Figure supplement 2—source data 1.** Related to *Figure 6—figure supplement 2*.

(*Figure 6—figure supplement 1*). These experiments exclude that the dominant effect of light stimulation on spasm generation originates from the activation of Vglut2$^+$ afferents since neither persistent firing of the Vglut2 terminals nor persistent dorsal root reflexes (*Bos et al., 2011*) were seen.

To further substantiate that glutamatergic spinal circuits support muscle spasms after chronic spinal cord transection, we performed optogenetic inactivation of eINs in *Vglut2$^{Cre}$; R26 eNpHR3-YFP* mice (*Hägglund et al., 2013*). Spasm generation was prevented when optogenetic silencing was simultaneously applied to the same segment as the electrical stimulation (*Figure 6DII–III* and *Figure 6—figure supplement 2D-E*) or when applied to the entire length of the isolated spinal cord (*Figure 6—figure supplement 2F*; compare with *Figure 6—figure supplement 2D*). This inhibitory effect was not due to inhibition of neurotransmitter release from all afferents since the monosynaptic inputs to motor neurons were intact during the co-incident electrical and light stimulation (compare insets in *Figure 6DII–III*) and the recorded dorsal-root-evoked potentials were reduced by less than 10% during light illumination (*Figure 6—figure supplement 2A–C*).

Finally, evidence that the spasms are the results of the recruitment of spinal eINs in several spinal segments comes from experiments where the optogenetic stimulation was delayed with respect to the electric stimulation in the same segment (n = 10, N = 3, *Figure 6D$_{IV}$*) or was restricted to distant segments from the electrically stimulated segment (n = 15, N = 3, *Figure 6—figure supplement 2G–H*). In these latter conditions silencing Vglut2$^+$ cells may phase delay the appearance of the spasm (*Figure 6—figure supplement 2G–H*; compare with *Figure 6—figure supplement 2D*) or reduce the duration of the spasm (*Figure 6—figure supplement 2G–H*; compare with *Figure 6—figure supplement 2D*) without completely blocking it.

These experiments show that activity of intrinsic excitatory neuronal circuits in the spinal cord is necessary and sufficient to trigger and maintain muscle spasms. The persistent neural activity is generated and maintained only in chronically but not in the acutely transected animals, highlighting the profound differences between the changes underlying spinal circuit reorganization in the chronic as compared to the acute phase of spinal cord transection.

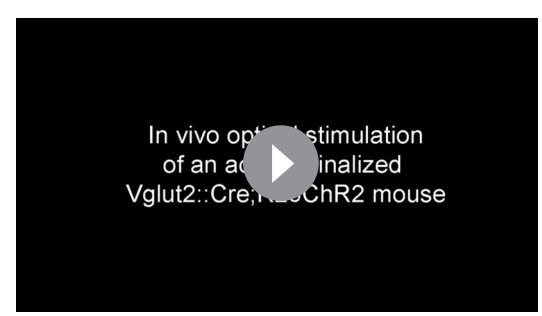

**Video 5.** Related to *Figure 6*. In vivo optogenetic activation of excitatory spinal interneurons triggers spasms in chronically spinal cord transected mice. The Movie shows a restrained *Vglut2$^{Cre}$; R26ChR2* mouse with the tail free to move. Short optogenetic stimulation (15 pulses of 20 ms duration at 10 Hz) of the first coccygeal segment of the spinal cord in acutely transected mice (control) generated a motor response during the stimulation. In contrast, a similar stimulation in chronically S2-transected mice induced a curling of the tail that lasted several minutes, similar to that seen during spontaneous or sensory-evoked spasms.

Recruitment of inter-segmentally connected Vglut2[+] neurons seems an essential network mechanism supporting spasm generation.

## Spinal iINs suppress motor responses after chronic spinal cord transection and do not generate the persistent neural activity

To test the possible contribution of iINs to muscle spasms generation, we selectively activate iINs by expressing ChR2 in VIAAT[+] cells (VIAAT[Cre]; R26ChR2-YFP). Optogenetic stimulation of iINs in vivo (n = 5) or in vitro (n = 7) never induced muscle spasms or sustained firing after chronic spinal cord transection (**Figure 6E_I**), demonstrating that activation of inhibitory synapses is not sufficient to excite MNs and initiate muscle spasms even though the chloride reversal potential might be shifted to more positive levels after SCI (**Boulenguez et al., 2010**).

Optogenetic in vivo activation of iINs in chronically spinal cord transected animals suppressed muscle spasms (**Video 6**, N = 4). However, the motor response suppression was related to the stimulation period and the spasms reappeared as soon as the stimulation terminated (**Video 6**). Similar results were obtained in the isolated spinal cord. When iINs were stimulated during the afferent activation it directly inhibited the synaptic release preventing development of a spasm (**Figure 6E_II**). When stimulating during muscle spasms these where prematurely inhibited (**Figure 6E_III**).

Lastly, light-induced inhibition of iINs during the pre-motor silent period phases advanced the prolonged motor neuron firing (**Figure 6F**) similar to in vitro experiments applying inhibitory blockers (**Figure 3G**).

The in vivo and in vitro experiments showed that iINs neither generate nor maintain muscle spasms. Rather inhibitory circuits generate functional inhibition onto motor neurons that sculpture the spasms generation by introducing a delay to their onset and participating in their termination.

## Discussion

Spasms after SCI reflect a general hyper-excitability of spinal circuits which in human and animal studies has been attributed to activation of motor neuron plateau potentials (**Bennett et al., 2001a**, **2001b**; **Eken et al., 1989**; **Hultborn et al., 2013**) and/or a reduction of inhibitory transmission (**Boulenguez et al., 2010**; **Crone et al., 1994**). Here we demonstrate that muscle spasms are generated by a combination of persistent activity in sequentially recruited spinal excitatory interneurons that interact with a graded expression of plateau potentials in motor neurons. Moreover, we show that inhibition is functional active and does not drive the spasms.

Several studies reported the appearance of long-lasting EPSPs at the motor neuron level after chronic spinal cord transection (**Akay et al., 2014**; **Brumovsky, 2013**) but the contribution of the excitatory inputs has been neglected as a direct cause of spasms generation and maintenance. By using a mouse model of chronic spinal cord transection that allows recordings from motor neurons as well as selective visualization and manipulation of identified groups of interneurons, we show the direct involvement of excitatory interneuron populations in generating the spasms. First, motor neurons receive a massive barrage of excitatory inputs during spasms. Second, the temporal dynamics of the excitatory spinal interneuron activity correspond to the spasm generation. Thirdly, light stimulation In Vglut2[Cre]; ChR2 mice initiated spasms. Although

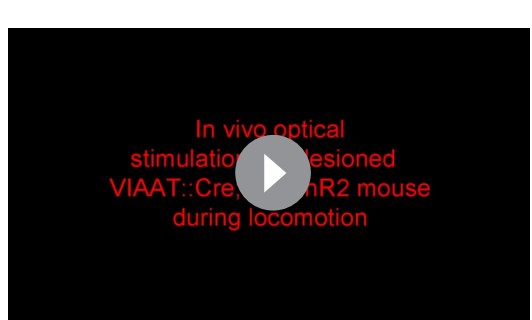

**Video 6.** Related to **Figure 6**. Optogenetic activation of spinal inhibitory neurons completely suppresses motor responses and muscle tone in chronically spinal cord transected mice. The Movie presents the bottom and lateral views of a chronically spinal cord transected mouse expressing ChR2 in inhibitory interneurons with a chronic implant for optogenetic stimulation over the first coccygeal segment of the spinal cord. Before light stimulation the tail was curved with a posture typical of chronic spinal cord transection. During the light activation of inhibitory interneurons, the tail completely relaxed and became flaccid. The effect ceased as soon as the optogenetic stimulation ended.

this stimulation may also activate terminals that express Vglut2 this activation was short-lasting, suggesting a substantial contribution from eINs activation to spasm generation. Finally, the absence or the severe reduction of spasms when eINs were optogenetically inhibited suggested a necessary role of eINs that could not be substituted by direct monosynaptic activation of motor neurons mediated by Vglut1 expression primary afferents (*Brumovsky, 2013*). All together these experiments suggest a pivotal role of eINs in triggering and sustaining the persistent neural activity. Importantly, the sustained activity was not generated by excitatory persistent dorsal root reflexes (*Bos et al., 2011*). Such prolonged effects were not elicited neither by electrical nor optogenetic stimulation in the present study.

With respect to motor neurons our study complement previous studies using a chronic spinal cord transection model of the sacral spinal cord in rats which showed a graded expression of plateau properties in motor neurons below the transection (*Bennett et al., 1999*). The plateau properties are generated by persistent $Ca^{2+}$ and $Na^+$ currents (*Li and Bennett, 2003*), whose manifestations are regulated by constitutively opened 5-HT receptors (*Murray et al., 2010*) or 5-HT released from cells in the spinal cord containing the enzyme aromatic L-amino acid decarboxylase (*Wienecke et al., 2014*). Our study also shows that motor plateau properties are more prevalent after chronic spinal cord transection, and the degree of expression of these properties matches the firing pattern of individual motor neurons during spasms. However, similar to what has been reported in the rat (*Bennett et al., 2001b*) we find that only about 10% of the motor neurons express full plateau potentials needed to support sustained firing (*Hounsgaard et al., 1988*; *Kiehn et al., 1996*) after chronic spinal cord transection. The majority of motor neurons expressed partial plateau properties, characterized by membrane properties that will promote but not by themselves sustain prolonged firing in the absence of synaptic inputs. Together, these findings underscore the contribution of these biophysical properties to prolonged firing in motor neurons after SCI but make it unlikely that motor neuron plateau properties are the main carrier of spasms.

Alteration in inhibitory circuits after SCI has been described in multiple studies as a result of reduced presynaptic inhibition (*Mukherjee and Chakravarty, 2010*; *Nielsen et al., 2007*), reduced reciprocal inhibition (*Mukherjee and Chakravarty, 2010*; *Nielsen et al., 2007*) and down-regulation of the KCC2 chloride-transporter in motor neurons (*Boulenguez et al., 2010*). In contrast, our findings explicitly demonstrate that the activity of iINs is increased after chronic spinal cord transection with no sign of a decreased efficacy of inhibitory inputs onto motor neurons. The presence of a silent period preceding electrically-evoked spasm suggested that inhibition was still activated by sensory stimulation and it was able to suppress motor neuron responses. Furthermore, the optogenetic activation of iINs in the sacral spinal cord was able to suppress spasms and all motor responses after chronic spinal cord transection. In contrast, iINs were not able to generate and/or maintain the persistent neural activity driving MNs during muscle spasms as revealed by optogenetic stimulation both in vivo and in vitro, showing that inhibition is not reverted to depolarizing IPSPs that triggers motor output. So the functional role of inhibition is to sculpture and terminate the prolonged spasms. Our study therefore shifts the focus away from a decreased inhibition as a cause of spasm generation and reassigns to iINs a key functional role in shaping the temporal expression of muscle spasms and its termination.

Profound changes at the network as well as at the single cell level appear in the chronic state of SCI in humans as well as in animal models (*Maier and Schwab, 2006*). Our study represents the first investigation where the functional impact of these changes has been directly tested, revealing insights about the operational logic of spinal circuits after injury. The appearance of persistent activity in spinal circuits may emerge from changes in network properties and/or biophysical properties of the spinal INs after transection. Indeed, the number of cells responding to low-threshold dorsal root stimulation appeared to be much higher after chronic spinal cord transection. Possible explanations for this are that proprioceptive afferents – whose activity is important for motor recovery after spinal cord transection (*Akay et al., 2014*; *Takeoka et al., 2014*) – provide stronger activation of excitatory neurons due to intraspinal sprouting of proprioceptive afferents (*Krenz and Weaver, 1998*) or due to reduced presynaptic inhibition (*Kathe et al., 2016*). The increased activity of excitatory interneurons may also engage mechanisms of feed-forward and/or recurrent excitation (*Abbinanti et al., 2012*; *Wang, 2001*). Feed-forward circuits among excitatory neurons may be recruited by the chronic spinal cord transection as a result of intraspinal sprouting. Moreover, since the populations of eINs were activated more than once, recurrent excitatory loops may develop or

be released after the transection. Finally, plateau potentials or voltage dependent amplification due to the presence of persistent inward currents in interneurons is also likely to be involved. Previous experiments have shown that interneurons in dorsal and ventral horns may express plateau properties before (*Abbinanti et al., 2012*; *Derjean et al., 2003*; *Russo and Hounsgaard, 1994*; *Theiss et al., 2007*) and after chronic spinal cord transection (*Dougherty and Hochman, 2008*; *Husch et al., 2012*). The mouse model offers future possibilities to selectively manipulate these properties in interneurons to understand the impact of such currents on the operation of the spinal network after SCI.

The clinical treatment of spasms relies mostly on drugs acting on GABA receptors suppressing motor responses at the premotor neuronal level (*Dietz, 2010*; *Li et al., 2004b*). The nature of the changes underlying chronic spinal cord transection, we described here, opens up new understandings and potential new targets in treating the maladaptive state of spinal circuits after SCI. Some of the most promising pharmacological approaches for the therapeutic management of traumatic SCI seem to exert their main motor effect through a silencing of glutamatergic transmission, like riluzole (*Krenz and Weaver, 1998*), and the different blockers of the serotonin receptors (*Murray et al., 2011*). Indeed selective interference with glutamatergic transmission could open the possibility to dampen the persistent activity in excitatory circuits, leaving the inhibitory ones intact, as obtained with drugs currently used for antiepileptic treatment and off target for chronic pain treatment (*Landmark, 2007*; *Sills, 2006*).

## Materials and methods

Wild-type mice of both sexes were used (4–6 months old, weight: 23 g ± 1.9, body length 8.2 cm ±0.3, mean ± SD). The following transgenic lines were used: $Vglut2^{Cre}$(*Borgius et al., 2010*), $VIAAT^{Cre}$ (*Hägglund et al., 2013*); *Rosa26-LSL-GCaMP3 (Ai38)*; *Rosa26-CAG-LSL- eNpHR3.0-EYFP-WPRE,* and *Rosa26-CAG-LSL-ChR2-EYFP-WPRE* (all from The Jackson Laboratories). All crosses were genotyped before experiments. Animals were housed in single cages an environmentally controlled room (22°C, 12:12 light-dark cycle). All surgical procedures and experimental manipulations were approved by the local ethical committee.

### Surgery for injury of the sacral spinal cord

The detailed surgical procedure for a complete transection of the sacral spinal cord of the mouse is described at Bio-protocol (*Bellardita et al., 2018*). All surgical procedures were performed under sterile conditions. Before surgery animals were deeply anesthetized with isoflurane and eye ointment was applied to protect the eyes form de-hydration. Deep anesthesia was confirmed when reflexes and tail pinch responses were absent. The second sacral segment (S2) was localized at the level of the second lumbar (L2) vertebral body (*Harrison et al., 2013*). The back of the animal was shaved and sodium iodine was applied to prevent infection. A vertical incision was made in the skin in over the spinous process of the L2 vertebral body and the muscles were cut with small eye scissor to localize the Ligamentum Flavum between the rostral part of the L2 vertebral body and the caudal part of the L1 vertebral body. Once the spinal cord was clearly visible, Xilocaine (1%) was applied locally on S2 before the tissue was aspirated with a small glass pipette (diameter of 100 μm), connected to a vacuum pump. Care was taken not to sever the dorsal and ventral arteries, which provide blood supply to the spinal cord below the transection. When the transection was visually seen to be complete, the muscles surrounding the spinal column were sutured to close the opening above the damaged spinal cord and to protect it from mechanical pressure. Then the skin was sutured and the animal was allowed to recover. Animals were given post-surgery treatment of Buprenorphine (0.1 mg/Kg) and Carprofen (5 mg/Kg) subcutaneously for 2 to 5 days. The injury only affects the tail muscles with no bladder or hindlimb involvement. However, animals were followed twice a day for the first week for signs of infection or change in body weight. Infections were treated with antibiotic and anti-inflammatory therapy (Tribrissen, subcutaneous for 5 days). Loss of more than 20% of body weight in the post-surgery period was indication to terminate the experiments. Only animals with a complete transection of the spinal cord, visually inspected during the dissection were included in the study.

## In vivo electromyography (EMG): recordings, stimulation, and analysis

For electromyography recordings the animals were restrained in a mouse restrainer with the tail hanging, free to move (see *Video 2*). Recording electrodes were inserted intramuscularly in the ventral part of the tail. Two types of electrodes were used: electrodes for gross EMG recordings and electrodes for single motor unit recording.

### Gross EMG recording electrodes

Two dual core Teflon-coated platinum/iridium wires with a diameter of 125 μm (WPI, code number PTT0502) were threaded through a 30 ½ gauge hypodermic needles. The Teflon cover was removed from about 0.5 mm of the tips and the two electrodes were then inserted 1–1.5 mm apart in the tail.

### Single unit recording electrodes

Two dual core Teflon-coated platinum/iridium wires with a diameter of 25 μm (WPI, code number PTT0110) were twisted around each other. They were threaded through a 30 ½ gauge hypodermic needle and were cut perpendicularly with a sharp scalpel.

For both gross EMG and single unit recordings a single wire of 125 μm, prepared like the electrodes for gross EMG recordings, was used as ground electrode and inserted at the level of the Co9 vertebral body. The proximal ends of the recording wires were connected to a differential amplifier (Custom made). The signal was band-pass filtered with 100–1000 Hz and sampled with 20 KHz for offline analysis.

### Stimulation

Tail nerves were stimulated by bipolar stimulation electrodes inserted into the tail using the same wire as for gross EMG recordings. Stimulation strength (constant current) was set to obtain EMG recordings that correspond in threshold and latencies to the 'M wave' and to the 'H reflex'. The M wave represents the direct stimulation of motor axons while the H reflex is generated by the monosynaptic activation of motor neurons by Ia afferent fibers. The amplitude of the H reflex is proportional to the number of activated motor neurons.

### Analysis of gross EMG and single unit recordings

Spike sorting and relative instantaneous frequency of the different motor units was obtained using Spike2 (CED products). Spikes were identified as being the same when a change in amplitude was less than 20% from the template waveform during the entire recordings and when all other parameters for template matching were unaltered. Motor units were divided based on the frequency distribution of the instantaneous firing frequency during at least five minutes of recording.

## In vitro sacral spinal cord preparation

The surgical procedure for isolating the sacral spinal cord of adult mice is described in more detail at Bio-protocol (*Bellardita et al., 2018*). To isolate the sacral spinal cord for in vitro experiments, mice were anesthetized with isoflurane and a laminectomy was performed caudal to the T12 vertebral body to expose the sacral spinal cord. Then the animal received pure oxygen for 5 min while the cord was continuously moistened with oxygenated modified artificial cerebrospinal fluid (in mM: 101 NaCl, 3.8 KCl, 18.7 $MgCl_2$, 1.3 $MgSO_4$, 1.2 $Kh_2PO_4$, 10 Hepes, 1 $CaCl_2$, 25 Glucose). Lastly the cord was quickly removed and placed in a perfusion chamber lined with Sylgard in normal Ringers solution that contained in mM: 111 NaCl, 3 KCl, 11 glucose, 25 $NaHCO_3$, 1.25 $MgSO_4$, 1.1 $KH_2PO_4$, 2.5 $CaCl_2$ oxygenated in 95% $O_2$ and 5% $CO_2$ to obtain a pH of 7.4 and maintained at 22–24°C. The preparation recovered for 1.5 hr before recording.

## In vitro stimulation and recordings

### Dorsal root stimulation

Dorsal roots were stimulated (usually at the first or second coccygeal segment) with glass pipettes attached to the dorsal roots. Single pulses or trains of 15 pulses of 50 μs duration were used. The stimulus strength is given as multiples of the threshold for the monosynaptically evoked response in the corresponding ventral root.

## Ventral root recordings

Ventral root activity was recorded with tight-fitting suction electrodes placed on various sacral and coccygeal ventral roots. The signal was filtered with 100–1000 Hz and sampled with 10 kHz.

## Motor neuron recordings

Sharp electrodes were made from thick-walled glass capillary tubes (1.5 mm OD; Warner Instrument G150F-4) using a Sutter P-97 micropipette puller (Sutter Instrument) and filled with 1 M K-acetate. Then the electrodes were beveled (BV-10 microelectrode Beveler, Sutter Instrument) until the resistance dropped to 25–30 M$\Omega$. Intracellular recordings of motor neurons were done in the discontinuous current clamp mode (DCC, switching rate 5–8 kHz, output bandwidth 5.0 KHz) using an Axoclamp 2B intracellular amplifier (Axon Instruments, Burlingame, CA). Data were sampled at 10 KHz. Only motor neurons with a stable resting potential below $-60$ mV and action potentials of at least 70 mV of amplitude were included in the study. Motor neurons were identified by their location (20 to 200 µm under the ventral surface of the spinal cord), by their distinctive action potential (AP)/afterdepolarization-afterhyperpolarization shape and by orthodromic conduction of action potentials in the corresponding ventral roots (see *Figure 3—figure supplement 1*). Motor neurons were stimulated with slow current ramps with standard durations of 8 s and amplitudes varying from 1 to 10 nA.

## Analysis of in vitro data

### Quantification of motor responses

We quantified the short latency reflex by averaging ventral root responses over a window 10 ms post stimulus and the long lasting responses over a time-window 50–4000 ms post stimulus, a period previously shown to reflect the motor unit activity during spasms. The responses were quantified using the integrated area of the rectified signal. The stimulus-curve response was graded as a multiple of the threshold intensity (T) for eliciting the smallest motor response.

### Estimation of plateau properties in motor neurons

The presence of plateau properties was estimated as: (1) difference in injected current ($\Delta$I) between recruitment ($I_{Re}$) and de-recruitment ($I_{De}$) during the ramps (expressed as $\Delta I = I_{De} - I_{Re}$), and (2) plotting the instantaneous frequency (calculated as the inter-spike intervals) against the injected current during the ramps.

### Estimation of the synaptic input to motor neurons

The change in the balance between the excitatory and inhibitory inputs onto motor neurons was estimated from the change in the number of excitatory postsynaptic potentials (EPSPs) and inhibitory postsynaptic potentials (IPSPs) before and after stimulation. Synaptic events where detected in time windows of 20 s before and 20 s after the electric stimulation of the dorsal root (train of 15 pulses of 50 µs duration at 10 Hz at 1T). Then the number of IPSPs and EPSPs were normalized using the formula:

$$\bullet \quad Inhibition = \frac{IPSPs}{IPSPs + EPSPs}$$

$$\bullet \quad Excitation = \frac{EPSPs}{IPSPs + EPSPs}$$

The previous normalization was applied to the IPSPs and EPSPs after stimulation as well.

## Calcium imaging

For visualizing activity in spinal interneurons we used crosses of *Rosa26-GAG-GCaMP3'* and *Vglut2-Cre* or *VIAAT^Cre* mice. The spinal cords were isolated from acutely or chronically spinal cord transected mice. The cord was transversally cut at the level of S4/Co1 and placed in a chamber mounted on an epifluorescence microscope (Zeiss, Axioscope2) with the dorsal and ventral horns facing the objective (10x or 40x). The cord was illuminated with a 100 W Mercury light source for excitation

(excitation filter, 470–490 nm) and visualization (emission filter, 520–560 nm). Activity-dependent changes in fluorescence were detected using a digital CMOS camera (Hamamatsu, Japan) at 10 and 20 frames/s and stored directly on the computer. Changes in fluorescence were analyzed off-line using an image processing software (ImageJ). Changes in fluorescence intensity over time, which reflect changes in the intracellular calcium concentration, were converted to $\Delta F/F = (Ft - Fo)/Fo \, x \, 100$ where $Ft$ is the fluorescence at any specific time $t$ and $Fo$ is the baseline fluorescence. Regions of interest (ROIs) were drawn over areas corresponding to dorsal and ventral horns or individual cells and $\Delta F/F$ variations over time were calculated for each ROIs.

## Normalization of calcium imaging data

To compare the activity of interneuron activity between preparations we applied a normalization procedure using the motor responses of the ventral root as a reference. Each considered ventral root response was divided in four periods and each period was normalized in 100 bins, preserving the temporal sequence of the response: the stimulation period was normalized with 100 bins between −1 and 0; the silent period with 100 bins between 0 and 1; the spasm with 100 bins between 1 and 2 and post-spasm silence with 100 bins between 2 and 3. Concurrently, the time of max activity (described as maximum averaged value, 500 ms, of $\Delta$F/F) for each IN was detected and normalized to the new value of the corresponding ventral root period. This normalization process was applied to all the INs considered in the study.

## Drugs

Picrotoxin (10 µM; Sigma), strychnine (1 µM; Sigma), mephenesin (1 mM; Sigma), and Kynurenic acid (0.5–1 mM; Sigma) were bath-applied.

## Immunohistochemistry, in situ hybridization and cellular counts

The animals were anesthetized with pentobarbital and perfused with 4% (w/v) paraformaldehyde (PFA) in Phosphate-Buffer-Saline (PBS). The sacral spinal cord was removed and post-fixed in 4% PFA overnight, rinsed in PBS, cryoprotected in 30% (w/v) sucrose in PBS overnight and cryo-embedded in OCT mounting medium. Transverse sections of the spinal cord (20 µm) were obtained using a cryostat. Sections were incubated for 24 hr at 4°C with the following primary antibodies diluted in PBS supplemented with 5% (vol/vol) fetal bovine serum and 0.5% Triton X-100: chicken anti-GFP (Aves Labs 1020; 1:2000), rabbit anti-RFP (1:1000, Rockland 600-401-379). Secondary antibodies were obtained from Jackson or Invitrogen and incubated at 1:400 for 4 hr at room temperature. A fluorescent Nissl staining (NeuroTrace Blue 1:200, Life Technology) was added during the secondary antibody incubation. Slides were rinsed and mounted in Vectashield medium and scanned on a LSM5 confocal microscope (Zeiss Microsystems) using x20 or x40 objectives. Multiple channels were scanned sequentially to prevent fluorescence bleed through and false-positive signals. A contrast enhancement and a noise reduction filter were applied in Adobe Photoshop for publication images. Fluorescent in situ hybridization combined with immunofluorescence labeling was performed as previously described (Borgius et al., 2010) using a Vglut2 probe spanning the base pairs 540–983 (produced by Dr. L. Borgius). Neurons were counted on the left and right side of the spinal cord slice, using raw z-stack confocal images obtained with a 20x objective and spanning the entire thickness of the sections. Counts were done manually with the help of the cell-counter plug-in in ImageJ, in three non-adjacent sections per mice (N = 3). Cellular counts per section were expressed as a percentage of glutamatergic cells (Vglut2 mRNA positive) expressing GCaMP3' per individual animal. A grand-mean ± standard deviation (SD) was calculated across animals to produce bar-graphs for the different strains.

## Optogenetic stimulation

Conditional transgenic mice expressing Channelrhodopsin2 or enhanced Halorhodpsin-3 were crossed with $Vglut2^{Cre}$ or $VIAAT^{Cre}$ mice (see Hägglund et al., 2013).

### In vivo experiments

Optical fibers for in vivo stimulation of spinal neurons were implanted as previously showed (Caggiano et al., 2014). Briefly, a laminectomy at the third lumbar spinal vertebra was performed to

expose the first segment of the coccygeal spinal cord. Small metal bars were used to clamp the vertebrae and kept in place by a mounting plate. The plate had a central opening over the spinal cord. The central opening was filled with silicone elastomer (World Precision Instruments). A cannula, maintained with a cannula holder on a stereotactic frame, was slowly lowered to 200 µm above the surface of the spinal cord at the level of the midline. The cannula was then secured by means of cyanoacrylate glue and dental acrylic and covered with a dust cap. The skin was sutured and the animal was allowed to recover. In vivo stimulation of spinal neurons was then performed connecting an optical fiber to the cannula with the animal: a) in a mouse restrainer (see previous section about EMGs) to evaluated and quantify the tail muscle responses with EMGs, or b) in a corridor with simultaneous high frequency camera acquisition (MotoRaterapparatus, TSE-Systems) to evaluate the overall behavioral response. Stimulations of ChR2 or eNpHR3 positive cells were obtained using a 473 nm or a 550 nm laser scanning system (UGA-40; Rapp Optoelectronic) triggered by Master 8, AMPI to deliver trains of stimuli (20 ms pulse duration, 10 Hz frequencies, and train durations of 1.5 s).

## In vitro experiments

Stimulation of the dorsal root and recording of dorsal and ventral roots were performed as previously described simultaneously with the optogenetic stimulation of ChR2 or eNpHR3 positive cells obtained with a 473 nm or a 550 nm laser scanning system (UGA-40; Rapp Optoelectronic) to illuminate with an intensity around 30 mW/mm$^2$ as measured with a laser power meter (Coherent) (*Bouvier et al., 2015*; *Hägglund et al., 2013*). The stimulation consisted of a train of 15 pulses (20 ms, 10 Hz).

## Data availability

Source data files accompanying *Figures 1–6* are available in XLSX file format. The raw data that support the findings of this study are available from the corresponding authors upon request.

## Acknowledgements

This work was supported by the Swedish Research Council, European Research Council (Locomotor-Integration), NINDS, and Hjärnfonden. We thank AC Westerdahl for genotyping and N Sleiers for animal breeding.

## Additional information

### Competing interests

OK: Reviewing editor, *eLife*. The other authors declare that no competing interests exist.

### Funding

| Funder | Grant reference number | Author |
| --- | --- | --- |
| European Research Council | 693038 | Ole Kiehn |
| National Institute of Neurological Disorders and Stroke | R01 NS090919 | Ole Kiehn |
| Hjärnfonden | | Ole Kiehn |
| Swedish Research Council | | Ole Kiehn |

The funders had no role in study design, data collection and interpretation, or the decision to submit the work for publication.

### Author contributions

CB, Conceived the study and designed all experiments, Developed the mouse model of chronic spinal cord transection, Performed the electrophysiological, calcium imaging and optogenetic in vitro experiments, Analysed all data, Wrote the paper; VCag, Performed the optogenetic in vivo experiments, Assisted with data analyses, Gave input in preparing the manuscript; RL, Performed the optogenetic in vivo experiments, Gave input in preparing the manuscript; VCal, Carried out fluorescent in situ hybridization experiments, Gave input in preparing the manuscript; AF, Contributed to the

electrophysiological experiments on motor neurons; AF, Gave input in preparing the manuscript; JB, Contributed to initial calcium imaging experiments, Gave input in preparing the manuscript; PL, Contributed to the anatomical experiments, Gave input for preparing the manuscript; OK, Conceived the study and designed all experiments, Assisted in analyses of data and interpretation, Wrote the paper, Supervise all aspects of the study

### Author ORCIDs
Ole Kiehn, http://orcid.org/0000-0002-5954-469X

### Ethics
Animal experimentation: All surgical procedures and experimental manipulations were approved by the local ethical committee and the Swedish Animal Welfare Agency and included in the ethical permit N. 29/2014.

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
