## [Decision Letter]

Thank you for submitting your article "Spatiotemporal correlation of spinal network dynamics underlying spasms after spinal cord injury" for consideration by *eLife*. Your article has been reviewed by three peer reviewers, and the evaluation has been overseen by a Reviewing Editor and Eve Marder as the Senior Editor. The following individuals involved in review of your submission have agreed to reveal their identity: Shawn Hochman (Reviewer #1); Robert M Brownstone (Reviewer #2); Aya Takeoka (Reviewer #3).

The reviewers have discussed the reviews with one another and the Reviewing Editor has drafted this decision to help you prepare a revised submission.

Summary:

The present work identifies a clear role for premotor neuronal networks in the generation of persistent motor activity generating muscle spasms after spinal cord transection. Insights provided profoundly alter our conceptual understanding of spinal cord transection-induced motor spasms and will undoubtedly lead to new avenues of exploration in the area of clinical therapeutics. The experimental work leverages a powerful sacral spinal cord/tail animal model of motor spasms with genetic crosses in transgenic mice to undertake high throughput assessment of the role of spinal excitatory (vGluT2) and inhibitory (VIAAT) interneurons and includes imaging of their activity with the genetically encoded calcium indicator (Cre-dependent GCaMP3) and their selective activation or inhibition with light activated CHR2 or Halorhodopsin. The work provides fundamentally novel insights at many levels into the neural phenomenon and putative neural circuits underlying spinal cord transection-induced motor spasms. The study is presented with clarity and does an exceptional job of parsing and relating temporal activity in inhibitory and excitatory interneurons to the expression of persistent motor activity. There are some concerns that should be addressed to make the conclusions of the paper strong and clarify some issues.

Essential revisions:

There are some concerns that should be addressed to make the conclusions of the paper strong and clarify some issues.

1) There are possible confounding issues with the experimental design that could impact interpretation/circuit dissection of events leading the generation of persistent motor activity. These are related to; (i) lack of consideration of genetic approaches in vGluT2 populations also including activation and or inhibition of afferents axons, (ii) lack of consideration of a contribution from spiking activity in afferents (e.g. evoked dorsal root reflexes initiated by actions from inhibitory interneurons), and (iii) lack of certainty that optogenetic actions on inhibitory interneurons is able to activate or inhibit sufficient population of interneurons generating the silent period. At the very least, these limitations need to be considered in the discussion.

One important control that should be performed is to show that optical excitation in vGluT2::CHR2 mice is only recruiting excitatory interneurons. Thus, in the Figure 6 panels using vglut2-cre mice, it is possible that optical illumination may also recruit (CHR2) or inhibit recruitment (halo) vGluT2^+^ primary afferents directly. One simple control experiment is undertaking the usual optical cord illumination in vgltu2::CHR2 mice but this time while recording in an adjacent dorsal root. This could be done in uninjured mouse without much added effort. Absence or presence in dorsal root recordings of time-locked antidromic spikes would determine whether they are or are not also directly activating intraspinal afferent axons. If they are, this needs to be integrated into the paper, because right now the authors are assuming the optical stimulus only activates (or inhibits) excitatory interneurons. If this experiment isn't to be be performed, the authors should justify why (e.g. perhaps these controls have already been done in other experiments using these mice).

The need for this control arises, for example in Figure 6D3, if some vGluT2^+^ primary afferents are recruited by the electrical stimulus, and they are also hyperpolarized by the optical stimulus, then pairing them could have prevented recruitment of this subpopulation of afferents. In this manner, lack of observed response may be not due to hyperpolarization of excitatory interneurons but to reduced recruitment of afferents initiating the response.

If the authors do such a control experiment and illuminate spinal cord as always in vGluT2::CHR2 mice and see no direct antidromic recruitment in recordings from the dorsal root, this would strongly support their untested presumption that optical actions in vglut2-cre mice result exclusively from changing the excitability of excitatory interneurons.

2) Some aspects of the presentation should be clarified and some of the conclusions scaled back a bit. A primary concern is the use of the term "SCI". The authors have done spinal cord transections – not "true" models of SCI (clip, drop weight, etc). They should be clear about using a spinal transection model (which is the appropriate model to use for the study), and refer to transection and not SCI throughout the manuscript (including the title).

Secondly, in the Abstract the authors say they "directly target all major classes of spinal interneurons" – there is no reason to say this. This led the reviewers to think that the authors were going to study each class independently. What they have done is to look at excitatory and inhibitory interneurons. The claim thus seems to overreach.

Thirdly, it is questionable about whether or not this is a "new model" – technically it is because it's in the mouse, but Bennett developed this model in the rat, and has been using it for decades. So it's more of an extension of a model. This claim of novelty should be scaled back.

3) The authors did not systematically follow spontaneous vs sensory induced spasm. Analyses were often skewed to sensory induced spasm and reasons were not given clearly why.

4) The authors should address whether spontaneous activity in primary afferents is responsible for the persistent activation of excitatory interneurons. This could be entertained as a possibility in the Discussion.

5) The authors should consider the potential connectivity rearrangements of primary afferents after injury may contribute to spasm.

6) The authors should address the possible contribution of "partial plateau potentials" in motoneurons to the spasms, and should be cautious about completely excluding these properties as playing a role.

---

## [Author Response]

*Essential revisions:*

*There are some concerns that should be addressed to make the conclusions of the paper strong and clarify some issues.*

*1) There are possible confounding issues with the experimental design that could impact interpretation/circuit dissection of events leading the generation of persistent motor activity. These are related to; (i) lack of consideration of genetic approaches in vGluT2 populations also including activation and or inhibition of afferents axons, (ii) lack of consideration of a contribution from spiking activity in afferents (e.g. evoked dorsal root reflexes initiated by actions from inhibitory interneurons), and (iii) lack of certainty that optogenetic actions on inhibitory interneurons is able to activate or inhibit sufficient population of interneurons generating the silent period. At the very least, these limitations need to be considered in the discussion.*

These are all valid points and we thank the reviewer(s) for bringing them up. We have addressed these issues in a number of ways including a more cautious writing, incorporating discussion of possibilities that we stimulate afferents, clarification of the text, and further experiments as outlined below and in response to individual reviewers comments.

*(i) lack of consideration of genetic approaches in vGluT2 populations also including activation and or inhibition of afferents axons,*

We recognize these issues. Using a transgenetic approach, as here, we will label Vglut2 expressing primary afferents that might also be stimulated by light. Experimentally we cannot completely exclude that possibility.

We have, however, done new experiments with recordings of the dorsal root potential during light stimulation. The experiments show that optogenetic stimulation of the dorsal root alone with a long pulse (1.5 second) generates a short-lasting antidromic activation of the dorsal root (about 30-50 ms long) despite a longer motor response appeared in the ventral root which in in acutely spinalized animals lasted as long as the stimulation and in chronically spinalized mice evoked a prolonged activity corresponding to a spasm (new Figure 5—figure supplement 1). These experiments exclude that a persistent activation of afferents cause the persistent activation of excitatory interneurons, inhibitory interneurons or motor neurons and also suggest that the light-activation of afferents may contribute but not be the main mechanism of spasms.

With regard to inhibition of Vglut2 expressing terminals with halorhodopsin, there are a number of observations from the present study that suggest this is not the primary cause of the reduction of the spasms.

1) The monosynaptic input to motor neurons elicited by electric stimulation of the dorsal root is the same with and without optogenetic inhibition of Vglut2 terminals (Figure 6DII-III insets), showing that there is still primary afferent excitation to the spinal network at least from the proprioceptive fibers.

2) We recorded the dorsal root during electric stimulation only and during simultaneous optogenetic inhibition of Vglut2 (as suggested by reviewer 1, new Figure 6—figure supplement 1). We see a less that 10% decrease in the evoked dorsal root responses due to optogenetic inhibition.

3) Moreover, optogenetic inhibition of Vglut2 cells can still inhibit development of spasms even when it is delivered after the termination of the stimulation (Figure 6CIV and Figure 6—figure supplement 1).

4) Lastly, if we electrically stimulate the dorsal root of a segment distant from the optogenetic stimulation (Figure 5—figure supplement 1D-G), we also see a complete suppression or a reduction of spasms, confirming that spasms are based on a polysynaptic spinal network involving Vglut2 cells.

(ii) lack of consideration of a contribution from spiking activity in afferents (e.g. evoked dorsal root reflexes initiated by actions from inhibitory interneurons), and

We have performed dorsal root recordings both in acutely and chronically spinal cord transected mice. We do see spike activity in the dorsal root after each stimulus, but it only last about 40 ms after in the beginning of the stimulus (Figure 5—figure supplement 1 and Figure 6—figure supplement 1) and never after this short-lasting activation. We are therefore convinced that prolonged firing of afferent fibers and/or dorsal root reflexes do not sustain the prolonged activity in excitatory neurons during spasms.

*(iii) lack of certainty that optogenetic actions on inhibitory interneurons is able to activate or inhibit sufficient population of interneurons generating the silent period.*

We do not have any direct evidence that we recruit the same number of inhibitory interneurons with electrical and optogenetic stimulation (and this is true for every study that use electric vs optogenetic approaches to drive cell firing). Presently, with the available transgenic approach we have no way of imaging (with GCaMP3’) and optogenetic stimulating (using ChR2) inhibitory neurons because imaging and stimulation use the same wavelengths. In small scale (less than 50 neurons) this can be done with viral approaches; experiments that are outside the scope of this study. The fact that optogenetic stimulation is working suggests that we activate a sizable number of inhibitory neurons. The fact that we inhibited inhibitory neurons responsible for the silent period is depicted in Figure 6 that shows phase advance of the spasm when the inhibition is suppressed in the silent period, similar to what happens when the inhibition is removed with drugs. We cannot think of any other experiments to show that more directly. Notable the activity of inhibitory neurons matches time-wise the silent period (Figure 5).

Both the monosynaptic input and the polysynaptic input during the stimulation is reduced, possibly through presynaptic inhibition of low threshold afferents. This is part of the effect that we want to show. Moreover, we also show that inhibition is active and able to terminate the ongoing spasm. We think that Figure 6 shows that clearly. Finally, activation of iINs does not trigger premature spasms.

*One important control that should be performed is to show that optical excitation in vGluT2::CHR2 mice is only recruiting excitatory interneurons. Thus, in the Figure 6 panels using vglut2-cre mice, it is possible that optical illumination may also recruit (CHR2) or inhibit recruitment (halo) vGluT2^+^ primary afferents directly. One simple control experiment is undertaking the usual optical cord illumination in vgltu2::CHR2 mice but this time while recording in an adjacent dorsal root. This could be done in uninjured mouse without much added effort. Absence or presence in dorsal root recordings of time-locked antidromic spikes would determine whether they are or are not also directly activating intraspinal afferent axons. If they are, this needs to be integrated into the paper, because right now the authors are assuming the optical stimulus only activates (or inhibits) excitatory interneurons. If this experiment isn't to be be performed, the authors should justify why (e.g. perhaps these controls have already been done in other experiments using these mice).*

As stated above we have now done these controls (new Figure 5—figure supplement 1 and Figure 6—figure supplement 1). There is activation of intraspinal afferent axons – but only for a very short period. The reverse experiments indicate a necessity for timed activity in eINs to activate the spasms (also supported by their activity and the recorded EPSCs/EPSPs in motor neurons during the spasm).

*The need for this control arises, for example in Figure 6D3, if some vGluT2^+^ primary afferents are recruited by the electrical stimulus, and they are also hyperpolarized by the optical stimulus, then pairing them could have prevented recruitment of this subpopulation of afferents. In this manner, lack of observed response may be not due to hyperpolarization of excitatory interneurons but to reduced recruitment of afferents initiating the response.*

*If the authors do such a control experiment and illuminate spinal cord as always in vGluT2::CHR2 mice and see no direct antidromic recruitment in recordings from the dorsal root, this would strongly support their untested presumption that optical actions in vglut2-cre mice result exclusively from changing the excitability of excitatory interneurons.*

The above points address these issues.

*2) Some aspects of the presentation should be clarified and some of the conclusions scaled back a bit. A primary concern is the use of the term "SCI". The authors have done spinal cord transections – not "true" models of SCI (clip, drop weight, etc). They should be clear about using a spinal transection model (which is the appropriate model to use for the study), and refer to transection and not SCI throughout the manuscript (including the title).*

We agree with this point and have changed the wording to transection that is more correct. We believe that the point raised here by the reviewers is extremely important in the field. We would like to point out thought that SCI is used many studies even after complete transection (e.g. Boulenguez et all., Nature Medicine, 2010 and Murray et all., Nature Medicine, 2010) and the field will definitely benefit from a clear definition of terms to indicate the specific type of lesions (transection vs spinal cord injury) as well as clear indication of the model (acute vs chronic and neonatal vs adult). This will help to specifically relate the interpretation of the results to the models used in the study.

*Secondly, in the Abstract the authors say they "directly target all major classes of spinal interneurons" – there is no reason to say this. This led the reviewers to think that the authors were going to study each class independently. What they have done is to look at excitatory and inhibitory interneurons. The claim thus seems to overreach.*

We have removed the word ‘all’.

*Thirdly, it is questionable about whether or not this is a "new model" – technically it is because it's in the mouse, but Bennett developed this model in the rat, and has been using it for decades. So it's more of an extension of a model. This claim of novelty should be scaled back.*

We do not completely understand this critique since we used the word only in the Abstract to indicate a “new” mouse model. Clearly Bennett generated the transection model of the sacral cord in the rat to study spasticity and plateau potentials, which was probably inspired by two studies: 1) the plateau potential work in Hultborn’s lab in cats after chronic spinal cord transection (Eken et al. 1999) and 2) the tail transection model first developed by Ritz et al. 1992 in cat.

However, we have removed the word ‘new’. We actually now (and in the previous version) explicitly give credit to the previous rat model in the Introduction: “To address these issues we have developed a chronic spinal cord mouse model that mimics aspects of SCI and offers the possibility to combine detailed electrophysiological and imaging studies of both motor neurons and neurotransmitter-defined populations of interneurons to directly monitor and perturb their activity during muscle spasms. The approach builds on an chronic spinal cord model developed in the rat (Bennett et al., 1999) that targets a transection to the sacral segment of the spinal cord thereby producing spasms in tail muscles without affecting limb muscles or bowel and bladder functions.”.

*3) The authors did not systematically follow spontaneous vs sensory induced spasm. Analyses were often skewed to sensory induced spasm and reasons were not given clearly why. Some of the Minor comments of Reviewer 3 address this issue.*

We mostly used the sensory induced spasms to investigate the neural circuits that generate them after chronic transection because sensory induced spasms allowed a better control for quantification of the observed effect. We go a long way to demonstrate that the spontaneous spasms in vivoand in vitroshare the same characteristics as the sensory evoked spasms and we have no reasons to believe that the underlying neuronal mechanisms should be different.

*4) The authors should address whether spontaneous activity in primary afferents is responsible for the persistent activation of excitatory interneurons. This could be entertained as a possibility in the Discussion.*

We have recorded from dorsal roots during sensory evoke spasms. We do not see maintained dorsal root activity. We now mention that in the text and have added a Figure 5—figure supplement 1.

*5) The authors should consider the potential connectivity rearrangements of primary afferents after injury may contribute to spasm.*

We aim at addressing these issues, which is not straight-forward, in future studies. However we introduce potential involvement of primary afferent reorganization in the Discussion:

“The appearance of persistent activity in spinal circuits may emerge from changes in network properties and/or biophysical properties of the spinal INs after transection. Indeed, the number of cells responding to low threshold dorsal root stimulation appeared to be much higher after chronic spinal cord transection. Possible explanations for this are that proprioceptive afferents – whose activity is important for motor recovery after spinal cord transection (Takeoaka et al. 2014; Turkay et al. 2014)- provide stronger activation of excitatory neurons due to intraspinal sprouting of proprioceptive afferents (Krenz and Weaver 1998) or due to reduced presynaptic inhibition (Kathe et al. 2016). The increased activity of excitatory interneurons may also engage mechanisms of feed-forward and/or recurrent excitation (Abbinanti et al., 2012; Wang, 2001). Feed-forward circuits among excitatory neurons may be recruited by the chronic spinal cord transection as a result of intraspinal sprouting. Moreover, since populations of eINs were activated more than once, recurrent excitatory loops may develop or be released after the transection. Finally, plateau potentials or voltage dependent amplification due to the presence of persistent inward currents in interneurons is also likely to be involved. Previous experiments have shown that interneurons in dorsal and ventral horns may express plateau properties before (Abbinanti et al., 2012; Derjean et al., 2003; Russo and Hounsgaard, 1994; Theiss et al., 2007) and after chronic spinal cord transection (Dougherty and Hochman, 2008; Husch et al., 2012). The mouse model offers future possibilities to selectively manipulate these properties in interneurons to understand the impact of such currents on the operation of the spinal network after SCI”.

*6) The authors should address the possible contribution of "partial plateau potentials" in motoneurons to the spasms, and should be cautious about completely excluding these properties as playing a role.*We agree with this comment and realize that we were too categorical in the previous version. We have qualified the description and specifically mention the contribution of the graded expression of these properties to spasms in the, Abstract, Results and Discussion. We retain, however, that the motor neuronplateau properties, although important, may play a less pronounced role in spasm generation than previously proposed. This does not exclude that plateau properties in interneurons play a role for spasm generation.